# Epigenetic Activation of TUSC3 Sensitizes Glioblastoma to Temozolomide Independent of MGMT Promoter Methylation Status

**DOI:** 10.3390/ijms242015179

**Published:** 2023-10-14

**Authors:** Qiong Wu, Anders E. Berglund, Robert J. Macaulay, Arnold B. Etame

**Affiliations:** 1Department of Neuro-Oncology, H. Lee Moffitt Cancer Center and Research Institute, 12902 Magnolia Drive, Tampa, FL 33612, USA; 2Department of Biostatistics and Bioinformatics, H. Lee Moffitt Cancer Center and Research Institute, 12902 Magnolia Drive, Tampa, FL 33612, USA; 3Department of Anatomic Pathology, H. Lee Moffitt Cancer Center and Research Institute, 12902 Magnolia Drive, Tampa, FL 33612, USA

**Keywords:** GBM, glioma stem cell, 5-Aza, TMZ, MGMT, methylation, TUSC3

## Abstract

Temozolomide (TMZ) is an important first-line treatment for glioblastoma (GBM), but there are limitations to TMZ response in terms of durability and dependence on the promoter methylation status of the DNA repair gene O^6^-methylguanine DNA methyltransferase (*MGMT*). MGMT-promoter-hypermethylated (MGMT-M) GBMs are more sensitive to TMZ than MGMT-promoter-hypomethylated (MGMT-UM) GBMs. Moreover, TMZ resistance is inevitable even in TMZ-sensitive MGMT-M GBMs. Hence, epigenetic reprogramming strategies are desperately needed in order to enhance TMZ response in both MGMT-M and MGMT-UM GBMs. In this study, we present novel evidence that the epigenetic reactivation of Tumor Suppressor Candidate 3 (TUSC3) can reprogram sensitivity of GBM stem cells (GSCs) to TMZ irrespective of MGMT promoter methylation status. Interrogation of TCGA patient GBM datasets confirmed TUSC3 promoter regulation of TUSC3 expression and also revealed a strong positive correlation between TUSC3 expression and GBM patient survival. Using a combination of loss-of-function, gain-of-function and rescue studies, we demonstrate that TUSC3 reactivation is associated with enhanced TMZ response in both MGMT-M and MGMT-UM GSCs. Further, we provide novel evidence that the demethylating agent 5-Azacitidine (5-Aza) reactivates TUSC3 expression in MGMT-M GSCs, whereas the combination of 5-Aza and MGMT inhibitor Lomeguatrib is necessary for TUSC3 reactivation in MGMT-UM GSCs. Lastly, we propose a pharmacological epigenetic reactivation strategy involving TUSC3 that leads to significantly prolonged survival in MGMT-M and MGMT-UM orthotopic GSCs models. Collectively, our findings provide a framework and rationale to further explore TUSC3-mediated epigenetic reprogramming strategies that could enhance TMZ sensitivity and outcomes in GBM. Mechanistic and translational evidence gained from such studies could contribute towards optimal design of impactful trials for MGMT-UM GBMs that currently do not have good treatment options.

## 1. Introduction

Glioblastoma (GBM) is the most common primary malignant tumor in adults. Due to its highly invasive and resistant characteristics, the multimodal standard-of-care therapies of surgery, radiation therapy (RT) and temozolomide (TMZ) afford a dismal median survival of only 15 months in newly diagnosed GBM patients [1,2]. TMZ demonstrates excellent anti-tumor activity against GBM, yet the duration of response in the adjuvant setting is limited secondary to therapeutic resistance [2]. Further, TMZ efficacy is also dependent upon the activity of O-6-methylguanine-DNA methyltransferase (MGMT) [2]. *MGMT* is a critical DNA damage repair gene that reverses the alkylating effects of TMZ and whose expression is regulated through promoter methylation [3,4,5,6]. Specifically, when the promoter is hypomethylated, MGMT expression is high. Conversely, hypermethylation of the MGMT promoter is associated with a low MGMT expression status. Accordingly, MGMT promoter methylation status is clinically used as a biomarker for TMZ response in GBM patients [7,8]. Clinically, MGMT-hypermethylated (MGMT-M) patients have suppressed MGMT protein expression, which leads to TMZ sensitization and prolonged survival [4,5,6]. In contrast, MGMT-hypomethylated (MGMT-UM) patients are resistant to TMZ and have much shorter survivals. When treated with TMZ and RT, MGMT-M GBMs have a median survival of 21.2 months versus only 14.0 months in MGMT-UM GBMs based on clinical trials [4,5,6]. So far, there have been no successful treatments to render MGMT-UM GBMs susceptible to TMZ, and therefore there is a desperate need for novel treatment strategies for MGMT-UM GBMs. Further, TMZ resistance is inevitable even in TMZ-sensitive MGMT-M GBMs. Hence, epigenetic reprogramming strategies are desperately needed in order to enhance TMZ response in both MGMT-M and MGMT-UM GBMs.

Several epigenetic modification mechanisms, including DNA methylation, histone methylation/acetylation, chromatin post-translational modification and non-coding RNAs modification, have been implicated in GBM malignant oncogenesis [9,10]. Reprogramming of such mechanisms could provide novel therapeutic opportunities to overcome TMZ resistance and improve GBM outcomes [11,12,13]. In particular, methylation alterations are prevalent in GBM, as already alluded to, and further promoter methylation status is an important determinant of TMZ sensitivity. The DNA methyltransferase enzymes that modulate promoter methylation therefore serve as ideal pharmacological targets. In this study, we evaluate how pharmacological targeting of DNA methyltransferase enzymes could be leveraged in overcoming epigenetic resistance in GBM mediated by GBM stem cells (GSCs).

There are currently several FDA-approved inhibitors of DNA methyltransferase enzymes, of which 5-Azacytidine (5-Aza) is the most well studied, with therapeutic activity against myeloid and solid tumors [14,15,16,17,18]. Mechanistically, 5-Aza inhibits DNA methylation through covalent binding as a nucleoside analogue to DNA methyltransferase 1 (DNMT1), leading to degradation of DNMT1. Further, there is growing evidence that 5-Aza has anti-proliferative activity against cancer-stem-cell-like subpopulations in breast and myeloid cancers [19,20]. In preclinical patient-derived glioma models, 5-Aza treatment was reported to inhibit tumor proliferation and induce cell differentiation only in tumors that harbor mutations in isocitrate dehydrogenase 1 (*IDH1-R132H*) gene [21,22]. We therefore sought to determine and target epigenetic mechanisms that mediate therapeutic synergies between 5-Aza and TMZ in GBM.

In this study, we provide evidence that 5-Aza synergizes with TMZ through epigenetic activation of Tumor Suppressor Candidate 3 (TUSC3) and that TUSC3 reactivation improves survival in MGMT-M and MGMT-UM GSCs. *TUSC3* is a tumor suppressor gene that is critical for proper N-glycosylation of nascent proteins in the endoplasmic reticulum [23,24,25,26]. TUSC3 has been reported to be downregulated in several cancers, including GBM [23,24,27,28]. Clinical study indicated a significant decrease in TUSC3 expression in glioma tissues compared with normal adjacent tissue, which is associated with higher malignancy [29]. In vitro study of GBM revealed that TUSC3 upregulation in GBM cell lines suppressed cell proliferation and invasion by inhibition of the Akt signaling pathway [30]. However, the precise role of TUSC3 in GSCs as well as details of the mechanisms remain unknown. Our study highlights the reprogramming role of TUSC3 activation in overcoming MGMT-mediated resistance and provides prospects for clinical translation through pharmacological epigenetic reactivation.

## 2. Results

### 2.1. GSCs Are Sensitized by 5-Aza to TMZ Treatment through Activation of TUSC3

Using 5-Aza as our DNMT1 inhibitor, we first tested 5-Aza inhibition efficiency by assessing DNMT1 protein expression and DNMT1 activity. The Western blot result indicated that 5-Aza successfully suppressed DNMT1 protein expression in both MGMT-M and MGMT-UM GSCs (Appendix A). Similar to Western blot, the measurement of DNMT1 activity demonstrated that DNMT1 enzyme activity was significantly suppressed by 5-Aza in all GSCs (Appendix A). Interestingly, we found that 5-Aza dramatically reduced cell viabilities in MGMT-M GSCs when treated in combination with TMZ, while there was no significant impact on MGMT-UM GSCs (Appendix A). This result suggested that 5-Aza may sensitize MGMT-M but not MGMT-UM GSCs to TMZ treatment. We further tested GSCs IC-50 against TMZ to validate the above observation. The results indicated that 5-Aza significantly decreased TMZ IC-50 in MGMT-M GSCs but had no impact on MGMT-UM GSCs (Figure 1A). Our findings suggested that 5-Aza could sensitize GSCs to TMZ treatment in an MGMT-promoter-methylation-status-dependent manner. Indeed, we observed a strong synergistic effect between TMZ and 5-Aza in MGMT-M GSC (Figure 1B, Loewe synergy score = 17.154). Supporting these findings, dual treatment with TMZ and 5-Aza also significantly suppress GSCs proliferation in MGMT-M GSCs compared to TMZ treatment alone (Appendix A). No significant difference in proliferation was found in MGMT-UM GSCs (Appendix A). Meanwhile, 5-Aza showed no impact on cell migration and invasion in either MGMT-M or MGMT-UM GSCs (Appendix A). To further investigate and validate the in vitro therapeutic synergies between TMZ and 5-Aza, we took advantage of orthotopic xenograft by intracranial implantation of GSCs. The animals were treated with vehicle control, TMZ only, 5-Aza only and TMZ combining 5-Aza. The results showed that 5-Aza alone did not significantly impact animal overall survival in animals bearing either MGMT-M (Figure 1C) or MGMT-UM (Appendix A) GSCs. However, the TMZ and 5-Aza combination significantly prolonged animal survival when bearing MGMT-M GSCs (Figure 1C), while the combination had no impact on the survival of animals bearing MGMT-UM GSC (Appendix A). Through our observation, it does appear that 5-Aza impacts GSCs sensitivity to TMZ treatment relative to MGMT promoter methylation status. However, 5-Aza had no influence either on MGMT mRNA expression in all the GSCs (Appendix A) or MGMT promoter methylation status (Appendix A). Since 5-Aza exhibited differential TMZ sensitivity reprogramming effects in MGMT-M versus MGMT-UM GSCs but no differential plasticity to MGMT status, we hypothesized that this phenomenon was due to epigenetic activation of specific gene(s) in MGMT-M GSCs by 5-Aza, which were not activated in MGMT-UM GSCs. We then performed microarray in MGMT-M (01 and 02) and MGMT-UM (01 and 02) GSCs treated with either DMSO control or 5-Aza to determine which gene(s) drove this phenomenon. According to the analysis, there were 42 genes that were activated by 5-Aza in MGMT-M GSCs but were not activated in MGMT-UM GSCs (Figure 1D, Red dotted circle). Further, we functionally screened these 42 gene candidates by siRNA silencing and viability assessment to determine which genes were closely associated with TMZ sensitization. The MGMT-M GSCs were treated with TMZ and 5-Aza and together with indicated siRNAs for all these 42 genes. We looked for specific gene(s) that could reverse the effects of combinatorial treatments with TMZ and 5-Aza. Notably, we found that 5-Aza failed to sensitize MGMT-M GSC to TMZ treatment only when the Tumor Suppressor Candidate 3 (*TUSC3*) gene was silenced, suggesting that *TUSC3* could be the critical gene that was activated during 5-Aza treatment and was responsible for TMZ sensitization (Figure 1E, Purple dotted rectangle). Evaluation of TUSC3 mRNA level in MGMT-M GSCs confirmed and further demonstrated that 5-Aza treatment could successfully activate TUSC3 expression in MGMT-M GSCs (Figure 1F).

### 2.2. TUSC3 Expression Is Epigenetically Regulated and Positively Correlated with GBM Overall Survival

We further looked into the clinical relevance of TUSC3 in GBM patients by analyzing the TCGA dataset. Full mRNA expression combined from microarray (Agilent) and RNA-seq (Illumina HiSeq RNASeq V2) was downloaded from https://gdc.cancer.gov/about-data/publications/lgggbm_2016 (accessed on 31 October 2022) and the TUSC3 expression level was extracted for the GBM samples [31]. Overall survival (OS) for the TCGA GBM samples was retrieved from the publication by Liu et al. [32]. A log-rank *p*-value was calculated, and median cut was applied in all survival analysis. The TCGA dataset analysis showed that high TUSC3 expression confers a significant survival benefit (Figure 2A, *p* = 0.0021). This survival benefit provides a strategy to efficaciously target GBM, taking advantage of TUSC3 epigenetic reactivation. Moreover, the TCGA GBM dataset analyses revealed a lower expression profile of TUSC3 in GBM patient tumors compared to normal brain tissues (Figure 2B). TCGA RNA-seq data were downloaded from https://gdc.cancer.gov/about-data/publications/pancanatlas (accessed on 31 October 2022) and log2-transformed and the *TUSC3* expression for the GBM samples was extracted. Interestingly, TUSC3 expression is negatively correlated with promoter methylation level, which confirmed that TUSC3 expression is epigenetically regulated in GBM (Figure 2B). To further confirm these observations, we have utilized 16 GBM-patient-derived samples to perform Illumina methylation 850 EPIC arrays (Illumina, San Diego, USA). TUSC3 had large variability in the methylation level across all samples and CpG-probes, but TUSC3 promoter was significantly highly methylated in the TSS1500 and TSS200 regions (Figure 2C).

### 2.3. Activation of TUSC3 Enhances TMZ Sensitivity in Both MGMT-M and MGMT-UM GSCs

To further investigate the role of TUSC3 in TMZ sensitization, we utilized wild-type (WT) and TUSC3-stable overexpressing GSCs (TUSC3-OE). TUSC3 overexpression in both MGMT-M and MGMT-UM GSCs was confirmed by Western blot for protein level and real-time qPCR for mRNA level (Figure 3A). We first evaluated IC-50 against TMZ with wild-type and TUSC3-OE GSCs. As shown, TUSC3 overexpression significantly decreased TMZ IC-50 values in both MGMT-M and MGMT-UM GSCs compared to wild-type (Figure 3B). Furthermore, TUSC3 overexpression significantly inhibited GSCs proliferation in both MGMT-M GSCs and MGMT-UM GSCs when treated with TMZ (Appendix A) compared to wild-type. These observations demonstrated that reactivation of TUSC3 could synergize with TMZ treatment in GSCs. We further established wild-type control (WT, with control sgRNA) and TUSC3 knockout (TUSC3-KO, with TUSC3 sgRNA) GSCs by CRISPR/Cas9 system in both MGMT-M and MGMT-UM GSCs (Figure 3C). We then performed a TUSC3 rescue experiment (with empty vector-EV and TUSC3 expressing construct) in wild-type and TUSC3 knockout GSCs (TUSC3-KO). Conversely, TUSC3 knockout led to an increase in TMZ IC-50 (Figure 3D), which was significantly reversed/downregulated by TUSC3 rescue (Figure 3D). This strongly suggested that 5-Aza relies on epigenetic reactivation of TUSC3 in order to sensitize GSCs to TMZ treatment either in MGMT-M or MGMT-UM GSCs.

### 2.4. Synergy of 5-Aza with MGMT Silencing to Influence TMZ Sensitivity in MGMT-UM GSCs

According to our observation, TUSC3 activation is critical for sensitizing GSCs to TMZ treatment. Since 5-Aza successfully activated TUSC3 in MGMT-M GSCs but not in MGMT-UM GSCs, we were interested in exploring strategies to activate TUSC3 in MGMT-UM GSCs. Our data suggested that 5-Aza could not sensitize MGMT-UM GSCs to TMZ due to high expression of MGMT secondary to hypomethylation of MGMT promoter. We therefore sought to determine whether 5-Aza could activate TUSC3 and sensitize MGMT-UM GSCs to TMZ if MGMT expression was suppressed. We hypothesized that 5-Aza and suppression of MGMT could be pharmacologically leveraged to reactivate TUSC3 expression in MGMT-UM GSCs and further improve TMZ treatment efficacy (Figure 4A). We first treated two different MGMT-UM GSCs (UM-01 and UM-02) with Lomeguatrib, an irreversible MGMT inhibitor that has no impact on MGMT promoter methylation status [33]. Lomeguatrib treatment inhibition efficiency was evaluated by Western blot for protein expression and qRT-PCR for mRNA expression. As shown, the inhibitor Lomeguatrib significantly decreased both MGMT protein (Figure 4B) and mRNA expression (Appendix A) in MGMT-UM GSCs. Importantly, assessment of TUSC3 mRNA expression indicated that combination of 5-Aza and Lomeguatrib successfully activated TUSC3 expression in MGMT-UM GSCs, while 5-Aza or Lomeguatrib alone had no impact on TUSC3 expression (Figure 4C). We then assessed whether the combination of 5-Aza and Lomeguatrib could further sensitize MGMT-UM to TMZ. Interestingly, the combination of 5-Aza and Lomeguatrib significantly decreased TMZ IC-50 in MGMT-UM GSCs compared to the insignificant impact of either inhibitor alone (Figure 4D), suggesting potential therapeutic synergy between 5-Aza and MGMT inhibitions in reactivating TUSC3. Surprisingly, not only in GSCs, we also made similar observations in GBM cell lines. Specifically, 5-Aza activated TUSC3 expression in U251 GBM cells (MGMT-hypermethylated) and further decreased TMZ IC-50 (Appendix A), while 5-Aza and Lomeguatrib combination successfully activated TUSC3 in T98G GBM cells (MGMT-hypomethylated) and sensitized cells to TMZ treatment (Appendix A).

### 2.5. Pharmacological Epigenetic Reactivation of TUSC3 Synergizes with TMZ, Leading to Positive Therapeutic Outcomes

To further quantify drug interactions between 5-Aza, Lomeguatrib and TMZ in MGMT-UM GSCs, we calculated drug combination index utilizing the three-way (5-Aza vs. Lomeguatrib vs. TMZ) drug interactions model. We selected five doses for each drug. The drugs were combined in a 3D grid, where each drug was linearly increased in one axis. All drug responses were determined by viability test. As shown in Figure 5A, three-drug combinations, TMZ, 5-Aza and Lomeguatrib, demonstrated potent synergistic effects in MGMT-UM GSC (ZIP synergy score = 22.562 ± 2.722). Strong synergistic effects of drug combination on GSCs should facilitate effective dosing and lower toxicity on normal neural cells (NSCs). We then further tested those three drug combination responses on GSCs and NSC with a serial of 2-fold dose escalation and found out that the effective dose (reached 90% response, green line) for GSC had a minor impact on NSC, which indicated low toxicity (Appendix A). The above observations provided further rationale for combinatorial synergy between 5-Aza and MGMT inhibitor with TMZ in the highly resistant MGMT-UM GSCs. Our studies have thus far shown that optimal combinatorial therapy with 5-Aza and MGMT inhibition (Lomeguatrib) had a significant therapeutic impact on MGMT-UM GSCs under TMZ treatment. Next, we wanted to validate the above findings in vivo, utilizing orthotopic xenograft models. Xenografts bearing MGMT-UM GSCs were established by intracranial injection. All the mice received TMZ treatment plus one of the following: (i) vehicle, (ii) 5-Aza, (iii) Lomeguatrib and (iv) 5-Aza combining Lomeguatrib treatment. The Kaplan–Meier survival curve indicated that the combination of 5-Aza and Lomeguatrib together with TMZ significantly prolonged mice survival, while 5-Aza alone with TMZ did not have significant influences (Figure 5B). MGMT and DNMT1 silencing efficiencies in solid tumors by 5-Aza and Lomeguatrib, respectively, were confirmed by Western blot (Figure 5C) and qRT-PCR (Figure 5D) with protein and mRNA extracted from tumor tissues. mRNA expression assessment also demonstrated that TUSC3 was successfully activated by 5-Aza and Lomeguatrib combination in vivo (Figure 5E). This result provided in vivo support that combining 5-Aza together with MGMT inhibitor and TMZ treatment could lead to significantly improved outcomes in MGMT-UM GBM.

## 3. Discussion

GBM is the most common and most lethal brain cancer, with a median survival of 14 months and a 5-year survival of less than 10% [2]. TMZ is an important first-line treatment for GBM, but TMZ response depends largely on the promoter methylation status of the DNA repair gene *MGMT*, which is a DNA repair enzyme that reverses the alkylating effects of TMZ by removing methyl groups from the O^6^-guanine position on DNA. Methylation of MGMT promoter (MGMT-M) suppresses MGMT protein expression, leading to TMZ sensitization. On the other hand, when the promoter is unmethylated (MGMT-UM), this facilitates MGMT protein expression and leads to TMZ resistance. The MGMT promoter is unmethylated (hypomethylated/MGMT-UM) in more than 50% of GBM patients. With the standard of care treatment paradigm, MGMT-M GBMs have a median survival of 21.2 months compared to 14.0 months in MGMT-UM GBMs. Strategies to epigenetically reprogram TMZ sensitivity of MGMT-UM GBMs to improve survival are desperately needed. Here, we have employed 5-Aza as an epigenetic reprogramming probe. It was previously reported that 5-Aza sensitized only *IDH1*-mutant gliomas [22]. Hence, it was previously hypothesized that 5-Aza sensitivity was governed by *IDH1* status. However, for the first time, we have made the seminal findings that MGMT methylation status governs GSCs plasticity to 5-Aza sensitization. We now report that 5-Aza actually sensitizes *IDH1*-wild-type tumors as well so long as MGMT activity is silent. It should be noted that *IDH1*-mutant tumors are most likely to have silent MGMT activity, which could explain the prior perception that *IDH1*-mutant status governed 5-Aza sensitization.

In this study, we found that 5-Aza was highly efficacious and specific in reprograming the TMZ sensitivity of MGMT-M GSCs both in vitro and in vivo, while the combination of 5-Aza together with Lomeguatrib was required in order to reprogram the TMZ sensitivity in MGMT-UM GSCs. Both 5-Aza and Lomeguatrib could penetrate the blood–brain barrier (BBB) and have been used in different clinical trials, thereby supporting translational feasibility [17,18,34,35,36,37,38]. From our results, it appears that the mechanistic underpinnings of TMZ sensitivity reprogramming involve epigenetic reactivation of TUSC3 expression. Interestingly, 5-Aza alone was sufficient in successfully reactivating TUSC3 in MGMT-M GSCs, while, in MGMT-UM GSCs, both 5-Aza and Lomeguatrib were required to reactivate TUSC3. Further, the drug response analysis of 5-Aza and Lomeguatrib on GSCs indicated that combination treatment with both inhibitors improved TMZ treatment sensitivity in MGMT-UM GSCs compared to either agent alone. Collectively, our findings have several interesting mechanistic and clinical implications.

It was surprising that, although 5-Aza is an established inhibitor of DNA methyltransferases, 5-Aza treatment did not significantly demethylate the hypermethylated MGMT promoter and sufficiently activate MGMT expression to mediate TMZ resistance in MGMT-M GBMs. Interestingly, 5-Aza treatment rendered MGMT-M GBMs sensitive and not resistant to TMZ, which would not be possible if 5-Aza had demethylated the MGMT promoter, leading to increased MGMT expression, which drives TMZ resistance. It is therefore likely that 5-Aza is not a significant inhibitor for the DNA methyltransferase that maintains the methylation status of MGMT. Hence, 5-Aza has no significant impact on MGMT promoter methylation status in GBM. Although 5-Aza has been reported to impact MGMT promoter in several cancers [39,40,41,42,43], DNA sequencing and quantification of mRNA and protein level revealed that MGMT activity was not always correlated with methylation of the core MGMT promoter [39]. Therefore, further studies are needed to address this phenomenon.

Another interesting finding was that TUSC3 transcriptional activation by 5-Aza was feasible only when MGMT activity was absent, as in the case of MGMT-M or MGMT-UM plus MGMT inhibitor Lomeguatrib. Our findings are supportive of the notion that MGMT may play a critical role in suppression of TUSC3 transcriptional activation through epigenetic regulation. Specifically, MGMT may inhibit demethylation of TUSC3 promoter when DNMT1 was suppressed by 5-Aza. Given that MGMT is not a known repressor of gene expression, it is possible that the suppression of TUSC3 promoter activation in MGMT-UM GSCs occurs through a complex or machinery mediated by MGMT, which suggests a novel role of MGMT in cancers. Meanwhile, it was noted that Lomeguatrib, as an MGMT inhibitor that works directly on MGMT protein, also decreased MGMT mRNA expression at specific time points. Interestingly, similar observations were reported and suggested that this phenomenon may be correlated with treatment time [44]. It was also reported that the decrease in mRNA level was transient. Further studies will be required to address the detailed molecular mechanisms underlying the impact of Lomeguatrib on MGMT mRNA.

A major clinical implication of our study is the translational epigenetic therapeutic synergy. Notably, in MGMT-UM GSCs, the combination of 5-Aza, Lomeguatrib and TMZ demonstrated the highest potency. Meanwhile, strong synergistic effects between TMZ and 5-Aza were observed in MGMT-M GSCs. Further, our work for the first time also shows that a three-drug combination (TMZ, 5-Aza and Lomeguatrib) demonstrated potent synergistic effects in MGMT-UM GSCs. Importantly, strong synergistic effects of drug combination on GSCs should facilitate effective dosing and lower toxicity on NSCs. The three-drug combination response showed a very minor impact on NSC compared with GSCs, suggesting the potential for low toxicity. Our observations support combinatorial synergy between 5-Aza, MGMT inhibitors and standard chemotherapy in GSCs, especially in the highly resistant MGMT-UM GSCs where MGMT expression is high. We have hereby devised a pharmacologic strategy for reactivating TUSC3 for reprogramming TMZ sensitivity in GBM, and we have demonstrated in vivo clinical translational feasibility with improved survival in MGMT-M and MGMT-UM GBM PDX mice models.

Given this finding that 5-Aza could increase TMZ sensitivity in MGMT-M GSCs but not MGMT-UM GSCs, our study provided further support for the hypothesis that this phenomenon is due to epigenetic activation of a specific gene in MGMT-M GSCs by 5-Aza, which was not activated in MGMT-UM GSCs. Accordingly, through expression array analysis, we determined that *TUSC3* was the driver gene for the above phenomenon, whereby TUSC3 expression was activated only in MGMT-M GSCs by 5-Aza. Failure of 5-Aza to fully sensitize MGMT-M GSC to TMZ treatment when gene *TUSC3* was silenced provides further support to the premise that *TUSC3* was the critical gene that was activated during 5-Aza treatment and that was responsible for TMZ sensitization. TCGA GBM dataset analyses revealed a lower expression profile of TUSC3 in GBM patient tumors compared to normal brain tissues. We subsequently demonstrated that TUSC3 expression is negatively correlated with promoter methylation level, which confirmed that TUSC3 expression is epigenetically regulated in GBM. Furthermore, clinical evidence of the overall survival showed that TUSC3 expression is positively correlated with patient outcome; specifically, high expression of TUSC3 significantly improves GBM patient survival, which was further confirmed by our Illumina methylation 850 EPIC arrays using 16 GBM-patient-derived samples. We further evaluated IC-50 against TMZ utilizing wild-type and TUSC3-stable overexpressed GSCs and demonstrated that TUSC3 overexpression significantly decreased TMZ IC-50 values in both MGMT-M and MGMT-UM GSCs compared to wild-type. Unfortunately, so far, little is known about how TUSC3 exerts a tumor suppressor gene function in GBM; therefore, further study regarding the TUSC3 downstream mechanism is worth pursuing in our future study. A major implication of our study is that epigenetic reactivation strategies can permit activation of tumor suppressor genes such as *TUSC3,* leading to therapeutic benefits. Traditionally in oncology, therapeutic efforts are often directed toward targeting and inhibiting oncogenic drivers. However, for the first time, to the best of our knowledge, we have successfully overcome the challenge of activating tumor suppressors through rational leverage of epigenetic agents.

In summary, we have identified a novel mechanism in GSCs involving TUSC3 that can be used to reprogram GSCs and further restore TMZ sensitivity and produce similar survival effects in both MGMT-M and MGMT-UM GSCs intracranial PDX models. Our study on pharmacologic epigenetic reprogramming with FDA-approved agents provides a rationale for translational feasibility and therefore will serve as a backdrop for a rational translational study in the future. Overall, these innovative studies will enlighten us on synergistic and resistant mechanisms associated with reactivation of TUSC3 to treat MGMT-UM GBMs. Our strategy has profound relevance for GBM patients and will provide a rationale for translational feasibility and insight into the novel epigenetic reprogramming mechanisms and contribute towards optimal design of impactful trials for MGMT-UM GBMs that do not have good therapeutic options.

## 4. Methods and Materials

### 4.1. Human Samples and Processing

GBM-patient-derived samples were collected from patients at Moffitt Cancer Center (MCC) (Tampa, FL, USA) and approved by Moffitt Cancer Center Institutional Review Board, and all patients signed consent forms that were approved by the Institutional Review Board. Fresh resected human brain tissue samples were from the operating room and taken within 2 h to the laboratory to start tissue dissection. Tissue specimens were washed in 1× PBS and dissected under sterile conditions. It was further processed with tumor dissociation process using the Liberase (MilliporeSigma, St. Louis, MO, USA). Afterward, the homogenate was filtered through a 40 µm cell strainer and centrifuged at 800× *g* rpm for 5 min to pellet the cells. Cells were then used for culture and downstream gDNA extraction and assays.

### 4.2. Cell Culture and Reagents

The patient-derived GSCs were well characterized (Table 1) and cultured in NS-A medium (90% NeuroCult NS-A Basal Medium Human plus 10% Human NeuroCult NS-A proliferation Supplements, StemCell Technologies, Vancouver, British Columbia, Canada in 3D sphere as we described before [45]. Complete medium was supplied with recombinant human epidermal growth factor (R&D System, Minneapolis, MN, USA) and 100 units/mL penicillin-100 μg/mL streptomycin (Life Technologies, Carlsbad, CA, USA). Anti-TUSC3 and anti-MGMT antibodies were obtained from Abcam (Cambridge, United Kingdom); anti-β-actin IgG-HRP was obtained from Santa Cruz Biotech (Dallas, TX, USA) and anti-DNMT1, Goat anti-Rabbit IgG-HRP and Goat anti-mouse IgG-HRP were obtained from CellSignaling Technology (Danvers, MA, USA). Temozolomide (TMZ) was obtained from Sigma (St. Louis, MO, USA). 5-Azacitidine (5-Aza) and Lomeguatrib were obtained from Selleckchem (Houston, TX, USA).

### 4.3. Gene Overexpression in GSCs

Human TUSC3 expression and empty vector constructs (bacterial stock) were obtained from Genscript (Piscataway, NJ, USA). Plasmid was purified using QuickLyse Miniprep Kit (Qiagen, Germantown, MD, USA). GSCs (2 × 10^5^) were seeded in 6-well plate 24 h before transfection. Lipofectmine 3000 (ThermoFisher Scientific, Waltham, MA, USA) was used for transfection following manufacture’s protocol. Puromycin (2 μg/mL, InvivoGen, San Diego, CA, USA) was used for selection after transfection and the puromycin concentration was then reduced to 1 μg/mL 5 days later for maintenance.

### 4.4. Gene Silencing by CRISPR/Cas9 System

CRISPR/Cas9 vectors lentiCRISPR-v2-puro was obtained from Addgene. sgRNA targeting human TUSC3 and sgRNA control were cloned into lentiCRISPR-v2-puro. TUSC3 targeting forward primer: CACCGCATTCGGAAGATTGAGCGTC, reverse primer: AAACGACGCTCAATCTTCCGAATGC; control sgRNA cloning forward primer: CACCGCACTCACATCGCTACATCA, reverse primer: AAACTGATGTAGCGATGTGAGTGC. Lentivirus was packed by 293T cells and cells were infected by lentivirus, followed by extensive selection of 2 μg/mL puromycin (InvivoGen, USA).

### 4.5. Western Blot Analysis

Heat-denatured proteins (50–100 µg) were loaded on 4–15% precast polyacrylamide gel (Bio-Rad, Hercules, CA, USA) along with molecular weight marker. The gel was run for 1–2 h at 120 V and then transferred to PVDF membranes (Bio-Rad) by semi-dry system. Membranes were blocked with 5% non-fat milk, 0.1% TBS-T (Tween) solution for 1 h at room temperature. The target proteins were then detected by the primary antibody at 4 °C overnight, washed with TBS-T (5 min each) and incubated with appropriate secondary antibody at room temperature for 1 h. The membrane was then washed three times with TBS-T, 5 min each. The target proteins were detected with luminol reagent and X-ray film (Santa Cruz Biotech, Dallas, TX, USA).

### 4.6. Real-Time PCR

Total RNA was extracted for cells or tissues using RNeasy mini-prep kit (Qiagen) and quantified with Nanodrop 2000 (ThermoFisher Scientific, Waltham, MA, USA). Real-time PCR was performed by the Bio-Rad CFX96 Touch Real-Time PCR Detection system using cDNA synthesized with the iScript cDNA Synthesis kit (Bio-Rad). Human MGMT and GAPDH primers were used as we described before [46]. Human DNMT1 forward primer: AAAGTCAGCATTTGCGGGTT, reverse primer: TACTTTACGTGGCCAAAGCG; human TUSC3 forward primer: GAGAGCTGATACTTTTGACCTCC, reverse primer: CCCGAATATGAACATCCGTTCTG.

### 4.7. Cell Viability Assay, Proliferation Curve and IC-50 Evaluation

Cell viabilities were measured by an XTT Cell Viability Assay Kit (Cell Signaling Technology, Danvers, MA, USA) following treatment/transfection of cells. The absorbance was measured at 450 nm using a microplate reader (Molecular Device, San Jose, CA, USA). For proliferation assay, 2000 cells were seeded in 96-well plate. Parallel cultured cells were counted and recorded every 2 days. Proliferation curve was carried out using GraphPad Prism. When tested IC-50 in GSCs and drug response in GSCs and NSCs, 5000 cells were seeded in 96-well plates. Drugs were added the next day in triplicates and treated for 72 h (5-Aza: 10 µM; Lomeguatrib: 10 µM). Following drug treatment, cell viability was assessed using XTT Cell Viability Assay Kit (Cell Signaling). Data analysis and calculation of IC-50 were carried out using GraphPad Prism 9 (Dotmatics, Boston, MA, USA).

### 4.8. DNMT1 Activity Assay

Nuclear extract was carried out according to the manufacturer’s instruction (Epigentek, New York, NY, USA). The activity of DNMT1 was detected by human DNMT1 ELISA kit according to the manufacturer’s instructions (Epigentek). The concentration of DNMT1 in samples was calculated by the working standard curve.

### 4.9. Trans-Well Migration and Invasion Assays

The trans-well migration and invasion assays were performed using 6-well Transwell^®^ Cell Migration and Invasion inserts according to the manufacturer’s instructions (Corning, New York, NY, USA). 1 × 10^5^ GSCs were seeded in trans-well and incubated 48 h (with DMSO or 5-Aza treatment) for migration and invasion. Migrating or invading cells were fixed with ethanol and stained with crystal violet. The numbers of migrated and invaded cells were quantified by counting under microscope.

### 4.10. Drug-Response Synergy Index Calculation

To quantify drug interactions between 5-Aza and TMZ and also to classify the interactions, we calculated combination index and isobologram. A cross design was made to test the synergy and sensitivity of a drug pair in MGMT-M GSCs. We simulated isobologram for the pair of 5-Aza and TMZ with five effective doses cross combinations 1/10 titrated (5-Aza from 10 µM, TMZ from 1000 µM) for 72 h. Loewe additivity model was used for two drugs combination. To further quantify drug interactions among 5-Aza, Lomeguatrib and TMZ, we calculated drug combination index utilizing three-way (5-Aza vs. Lomeguatrib vs. TMZ in MGMT-M GSCs) drug interactions model. We chose five doses for each drug treatment with indicated titration for 72 h. For TMZ, the dose was titrated from 330 µM, while 5-Aza, Lomeguatrib were titrated from 33 µM and 20 µM. Drugs were combined in a 3D grid, where each drug was linearly increased in one axis. ZIP model was used for three drugs combination. All drug responses were determined by viability test. For synergy score measurement, data matrix was input and analyzed by SynergyFinder v3.0 (https://synergyfinder.fimm.fi).

### 4.11. Genomic DNA Extraction and Bisulfite DNA Sequencing Analysis

Genomic DNA (gDNA) was extracted from GSCs using Quick-DNA Miniprep kit (Zymo, Irvine, CA, USA) and bisulfite converted using EpiTect Fast DNA Bisulfite Kit (Qiagen, Germantown, MD, USA) according to the manufacturer’s protocol. The bisulfite-converted DNA was subjected to PCR amplification using specific primers of MGMT promoter to amplify a 310 bps region of the promoter CpG island [46], forward: TTGAGTTAGGTTTTGGTAGTGTTTAG, reverse: CCTTTTCCTATCACAAAAATAATCC. PCR products were purified and cloned using pGEM-T Easy Vector System (Promega, Madison, WI, USA). DNA from positive clones were extracted by Qiaprep Spin Miniprep Kit (Qiagen) for sequencing.

### 4.12. Microarray

Total RNA was extracted using RNeasy mini-prep kit (Qiagen) and followed by DNase digestion using RNase-Free DNase Set (Qiagen). Extracted RNA was labeled and hybridized onto the GeneChip PrimeView Human gene expression array cartriade (Affymetrix, Santa Clara, CA, USA).

### 4.13. siRNA Knockdown

GSCs were transfected with small interfering RNA control (MilliporeSigma) or indicated gene siRNA (MilliporeSigma) using Lipofectamine RNAiMAX Transfection Reagent (ThermoFisher Scientific). One day prior to transfection, the cells were seeded in 24-well plate (5 × 10^4^) and siRNAs (10 pmol per well) were prepared according to the manufacturer’s instructions and added to the cells. Cells were cultured 48 h and then for further assays.

### 4.14. Rescue Experiments

Rescue experiments were performed in wild-type and TUSC3 knockout GSCs. In rescue groups, these GSCs were transiently transfected with either empty vector or full-length TUSC3 cDNA. Transfection was followed manufacturer’s instructions (Lipofectmine 3000, ThermoFisher Scientific).

### 4.15. Mice and GBM Xenograft Model

Further, 6–8 weeks female NCRNU athymic mice were purchased from Taconic Biosciences. All animals were housed in the American Association for Laboratory Animal Care–accredited Animal Resource Center at Moffitt Cancer Center. All animal procedures and experiments were carried out under protocols approved by the Institutional Animal Care and Use Committee of the University of South Florida and Moffitt Cancer Center. All animal studies were performed in accordance with relevant guidelines and regulations of University of South Florida and Moffitt Cancer Center. For orthotopic model, xenograft tumors were established by intracranially injecting 1 × 105 indicated GSCs in a 2–3 μL volume of PBS in the right striatum of mice (n = 5/group) on a Stoelting Digital Stereotaxic Instrument (Stoelting, Wood Dale, IL, USA). Seven days after implantation, the mice were randomized into distinct groups for treatment (as indicated in figures). TMZ, 50 mg/kg/day; 5-Aza, 5 mg/kg/day; Lomeguatrib, 20 mg/kg/day. For survival studies, animals were followed every day until they lost 20% of body weight or had trouble ambulating, feeding or grooming.

### 4.16. Illumina Methylation 850 EPIC Arrays

Total genomic DNA was extracted using the Quick-DNA Miniprep kit (Zymo, Irvine, CA, USA) and followed by RNase A digestion. A DNA methylation assay using an Illumina DNA methylation MethylationEPIC array (Illumina, San Diego, USA) was performed on genomic DNA extracted from 16 GBM patient tissues. Minfi was used for normalization using Noob and FunNorm functions [47,48,49].

### 4.17. Bioinformatics Analysis

In-house gene expression data. The Affymetrix U133 plus 2.0 CEL files were normalized using IRON and log2-transformed before analysis [50]. The following criteria were used for selecting the most important genes: the fold change (FC) between the naïve (DMSO-treated) and the 5-Aza-treated cells was calculated. Probe sets with FC  >  1 for both the MGMT-M and MGMT-UM, max (expression value naïve, 5-Aza treated)  >  4 and matching signs were selected. Probe sets mapping to multiple genes or un-annotated were discarded. The final gene-level list was derived by selecting a single probe set for each gene.

TCGA GBM TUSC3 expression vs. Methylation. The GBM methylation data were downloaded as raw IDAT files and normalized via internal control probes followed by background subtraction. 4 *TUSC3* CpG-probes located in the TSS200-5′UTR with r < −0.5 were averaged and correlated to *TUSC3* expression level.

TCGA GBM survival analysis. Bioinformatics analysis and figures were completed using MATLAB 9.8 and Statistics and Machine Learning Toolbox 11.7, The MathWorks, Inc., Natick, Massachusetts, United States. Survival analysis was completed in MatSurv v1.1.03 [51].

### 4.18. Statistics

Student’s *t*-test (for two-condition experiments) and ANOVA (for multiple-condition experiments) were employed. Survival was assessed using Kaplan–Meier analysis with statistical comparisons made by log-rank (Mantel–Cox) test. All statistical tests were considered significant at *p* < 0.05. * Means *p* < 0.05. For survival data, a sample size of 5 mice per group to obtain 80% power to detect a hazard ratio of 0.22 (control versus treatment group) at 5% type I error by a two-sided log-rank test.

## Figures and Tables

**Figure 1 ijms-24-15179-f001:**
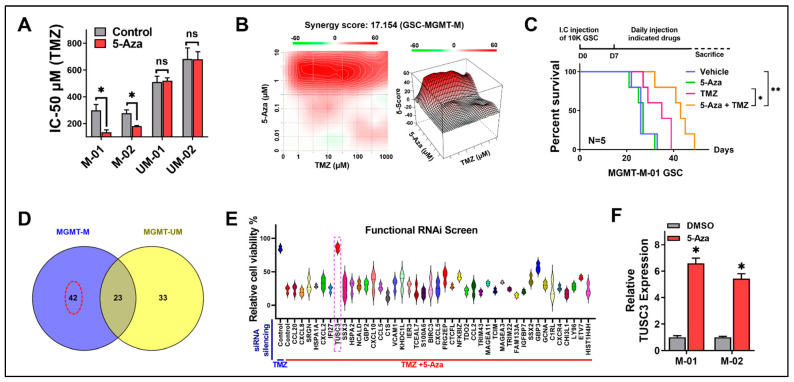
5-Aza sensitizes GSCs to TMZ treatment through activation of TUSC3. (**A**) TMZ IC-50 alteration between DMSO- and 5-Aza-treated GSCs. *n* = 3; *, *p* < 0.05; ns, no significance. (**B**) Synergy index of agents 5-Aza and TMZ in MGMT-M GSC were analyzed by SynergyFinder. Synergy sore = 17.154. (**C**) Kaplan–Meier survival curves of immunocompromised mice bearing MGMT-M GSCs treated with Vehicle, 5-Aza only, TMZ only or 5-Aza plus TMZ. The *p*-values were calculated by Mantel–Cox log-rank test. *, *p* < 0.05; **, *p* < 0.01. I.C., intracranial. (**D**) Venn diagram of microarray cross analysis of microarray of gene alternation by DMSO control and 5-Aza treatment in MGMT-M and MGMT-UM GSCs. Red dotted circle indicates selected 42 genes. (**E**) Functional siRNA screen with selected 42 gene candidates by cell viability assessment. Cells were treated with TMZ alone or TMZ plus 5-Aza. TUSC3 was selected as our target that indicated by purple dotted rectangle. (**F**) mRNA expression of TUSC3 was measured by qPCR in MGMT-M GSCs. Data are presented as mean ± SD. *, compared to DMSO control. *n* = 3; *, *p* < 0.05.

**Figure 2 ijms-24-15179-f002:**
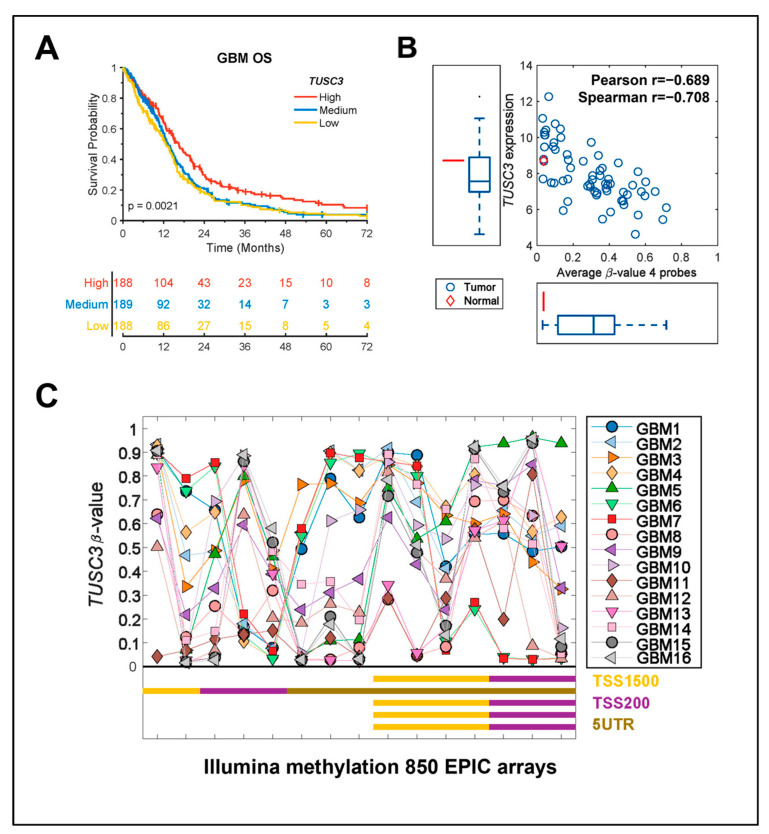
TUSC3 expression is epigenetically regulated and positively correlated with GBM overall survival. (**A**) Kaplan–Meier survival plots of TUSC3 expression from TCGA GBM dataset analysis. The survival plots include high, medium and low TUSC3 expression. *p* = 0021; high vs. low. (**B**) Correlation of TUSC3 expression and promoter methylation in normal brain tissue and GBM patient tumor with TCGA dataset analysis. Red diamond as normal brain tissue; blue circle as GBM patient tumor. X axis: average β-value, 4 probes; Y axis: *TUSC3* expression. Pearson r = −0.689; spearman r = −0.708. (**C**) Illumina methylation 850 EPIC arrays analysis of 16 GBM-patient-derived samples for TUSC3 methylation status with probes including TSS1500, TSS200, 5′UTR. The *Y*-axis shows the level of methylation ranging from zero to one. The *X*-axis shows the probe Id position and genomic position for all the probes. The different colored shaped spots indicate CpG islands and gene body type as described in the figure.

**Figure 3 ijms-24-15179-f003:**
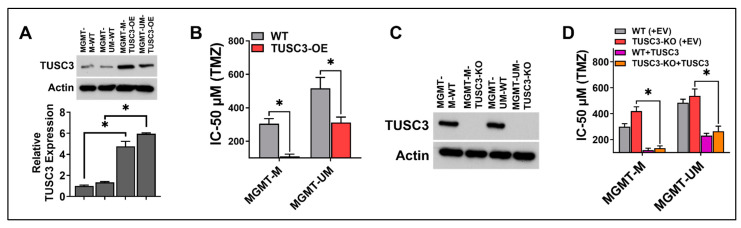
Activation of TUSC3 enhances TMZ sensitivity in both MGMT-M and MGMT-UM GSCs. (**A**) Immunoblotting and qPCR validation of TUSC3 overexpression in GSCs. (**B**) TMZ IC-50 alteration in wild-type control and TUSC3-overexpressed GSCs. (**C**) Knockout efficiency of TUSC3 examined by immunoblotting in MGMT-M and MGMT-UM GSCs. (**D**) TMZ IC-50 alteration in TUSC3 rescue experiment (transient TUSC3 overexpression). EV, empty vector; WT, wild-type control; TUSC3-KO, TUSC3 knockout GSCs. Data are presented as mean ± SD. *n* = 3; *, *p* < 0.05.

**Figure 4 ijms-24-15179-f004:**
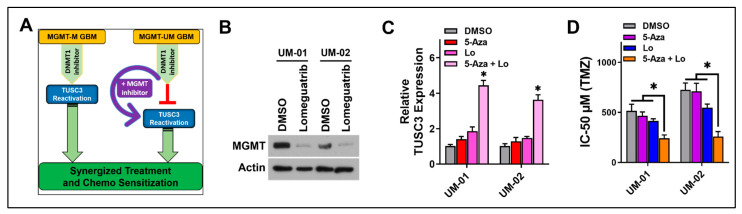
5-Aza synergizes with MGMT silencing to influence TMZ sensitivity in MGMT-UM GSCs. (**A**) Schematic diagram of TUSC3-activation-centered pharmacological therapeutic strategy through combination of DNMT1 inhibition, MGMT inhibition and TMZ. (**B**) Immunoblotting of MGMT protein in MGMT-UM GSCs (01 and 02) to examine Lomeguatrib efficiency of MGMT inhibition. β-actin was used as loading control. (**C**) mRNA expression of TUSC3 was measured by qPCR in MGMT-UM GSCs with indicated single or combination treatment for 72 h. (**D**) TMZ IC-50 alteration during indicated treatment (4 treatment groups) in MGMT-UM GSCs. Lo, Lomeguatrib. Data are presented as mean ± SD. *n* = 3; *, *p* < 0.05.

**Figure 5 ijms-24-15179-f005:**
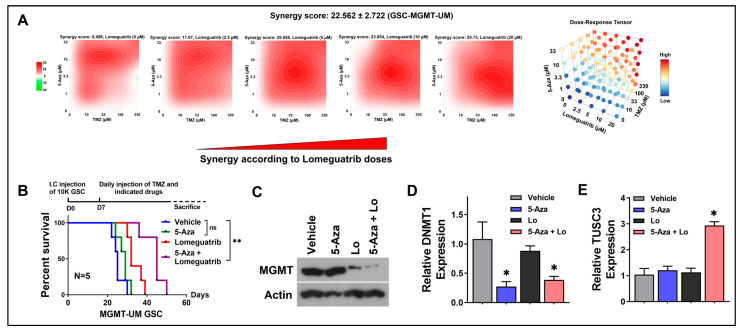
Pharmacological epigenetic reactivation of TUSC3 synergizes with TMZ, leading to positive therapeutic outcomes. (**A**) Synergy index of 5-Aza, TMZ and Lomeguatrib in MGMT-UM GSC analyzed by SynergyFinder. Synergy score = 22.562 ± 2.722. (**B**) Kaplan–Meier survival curves of immunocompromised mice bearing MGMT-UM GSCs. The mice were randomly assigned to 4 treatment groups, all with TMZ, and respectively combining (i) vehicle, (ii) 5-Aza, (iii) Lomeguatrib or (iv) 5-Aza plus Lomeguatrib. The *p*-values were calculated by Mantel–Cox log-rank test. **, *p* < 0.01; ns, no significance. I.C., intracranial. (**C**) Immunoblotting of MGMT protein to examine Lomeguatrib efficiency of MGMT inhibition. β-actin was used as loading control. (**D**,**E**) mRNA level of DNMT1 and TUSC3 measured by qPCR in tumors from indicated treatment groups. Lo, Lomeguatrib. Data are presented as mean ± SD. *n* = 3; *, *p* < 0.05.

**Table 1 ijms-24-15179-t001:** GSCs characterization.

ID	Growth	SOX2	CD133	MGMT	Nestin
MGMT-M-01	Spheres	+	+	hypermethylated	+
MGMT-M-02	Spheres	+	+	hypermethylated	+
MGMT-UM-01	Spheres	+	+	hypomethylated	+
MGMT-UM-02	Spheres	+	+	hypomethylated	+

+: positive expression.

## Data Availability

All the datasets analyzed during the current study are provided by links in Results and Methods. The data that support the findings of this study are stored at Moffitt Cancer Center and Research Institute and are available from the corresponding authors upon reasonable request if required.

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
