# Peer review of "Epigenetic Activation of TUSC3 Sensitizes Glioblastoma to Temozolomide Independent of MGMT Promoter Methylation Status"

_ijms, 2023, doi:10.3390/ijms242015179_

Round 1
Reviewer 1 Report
The paper by Wu et. al. addresses a crucial need to find a new effective therapy for patients with GBM, especially with TMZ-insensitive GBM. It's a very well-designed and written manuscript. The discussion is well-led and easy to understand. However, I'd like to provide a few comments:
1) I'd like to suggest changing the color schema of the images to make them accessible for color-blind readers (avoid blue, red, and green in close proximity) and make them more cohesive.
2) Figure 2. B is missing an axis description. It's tough to see the correlation between the methylation of the promotor and the expression of TUSC3.
3) Lines 232-240: Authors indicate that lomeguatrib treatment decreased expression of both MGMT protein and mRNA (Fig 4.B-C). Since this inhibitor works directly on the MGMT protein, can it change the expression profile of mRNA? I think it needs to be discussed in depth.
4) Paper describes the potential therapeutic approach to asses the TMZ-resistance problem. It'd be beneficial to discuss pathways and correlations between the expression of TUSC3 and MGMT.
5) In Material & Methods the experiment of knockdown cannot be repetitied without providing more details on amounts of used siRNAs. A detailed description is recommended for the Western Blot procedure.
Author Response
The paper by Wu et. al. addresses a crucial need to find a new effective therapy for patients with GBM, especially with TMZ-insensitive GBM. It's a very well-designed and written manuscript. The discussion is well-led and easy to understand.
R: Thanks very much for the reviewer’s comments and we have addressed the questions and concerns point by point below.
1. I'd like to suggest changing the color schema of the images to make them accessible for color-blind readers (avoid blue, red, and green in close proximity) and make them more cohesive.
R: We thank the reviewer for this great suggestion. We have changed the color schemes of some of the images so that may be easier for color-blind readers.
2. Figure 2. B is missing an axis description. It's tough to see the correlation between the methylation of the promotor and the expression of TUSC3.
R: We thank the reviewer for the comments. In Fig.2B, the X-axis is Average β-value 4 probes (with the mean value column below) and the Y-axis is TUSC3 expression (with the mean value column left). For additional clarity, we have included the axes description in the “Figure legend” of Fig2.
3. Lines 232-240: Authors indicate that lomeguatrib treatment decreased expression of both MGMT protein and mRNA (Fig 4.B-C). Since this inhibitor works directly on the MGMT protein, can it change the expression profile of mRNA? I think it needs to be discussed in depth.
R: We thank the reviewer for the suggestion. Yes, we agree with the reviewer, that the MGMT inhibitor Lomeguatrib does work directly on MGMT protein and activity, which was shown in Fig. 4B. Lomeguatrib is a pseudosubstrate that binds to MGMT leading to rapid degradation of MGMT via ubiquitination. In our study, we assessed the impact of Lomeguatrib with the MGMT mRNA expression measurement (Fig.S7) given prior reports by Cioccoloni et al 2015 – see reference 44 – that Lomeguatrib treatment could transiently decrease MGMT mRNA levels. Accordingly, MGMT mRNA expression was decreased when we treated GSCs with Lomeguatrib. Since we only tested this at 72 hours, it may vary if we test at different time points. Since the effects of Lomeguatrib on MGMT mRNA effects have been previously reported as transient, the relevance here is limited. Our focus is mainly on MGMT protein. We will, therefore, only use protein level as Lomeguatrib treatment indicator. We also have discussed the above in the “Discussion” section line 354-360.
4. Paper describes the potential therapeutic approach to assess the TMZ-resistance problem. It'd be beneficial to discuss pathways and correlations between the expression of TUSC3andMGMT.
R: We appreciate the reviewer for this great suggestion. In our study, 5-Aza alone was sufficient in successfully reactivating TUSC3 in MGMT-M GSCs, while in MGMT-UM GSCs both 5-Aza and Lomeguatrib were required to reactivate TUSC3. This observation strongly suggests that MGMT itself may play a critical role in transcriptional suppression of TUSC3. Given that MGMT is not a known repressor of gene expression, it is possible that the suppression of TUSC3 promoter activation in MGMT-UM GSCs occurs through a complex or machinery that mediated by MGMT, which suggests a novel role of MGMT in cancers. We added those discussions in the “Discussion” section now, line 346-354.
5. In Material & Methods the experiment of knockdown cannot be repetitied without providing more details on amounts of used siRNAs. A detailed description is recommended for the Western Blot procedure.
R: We thank the reviewer for this suggestion. Now we added the usage of siRNAs, and detailed the procedure of western-blot.
Reviewer 2 Report
This manuscript entitled "Epigenetic activation of TUSC3 sensitizes glioblastoma to temozolomide independent of MGMT promoter methylation status." shows that treatment with 5-Aza sensitized to TMZ via upregulation of TUSC3 in MGMT-methylated glioma stem cells, and combination of 5-Aza and a MGMT inhibitor sensitized to TMZ in MGMT-unmethylated GSCs. This article is of value in the point that it may be provide a novel adjuvant treatment modality for TMZ-resistant glioblastoma. This reviewer has two concerns before recommendation for publication.
(1) Figure 3
Because 5-Aza is a DNA methylation inhibitor, and MGMT is also influenced by its treatment, immunoblots of MGMT should be indicated in Fig 3A and 3C.
(2) GSCs were used in experiments of 5-Aza treatments with/without Lomeguatrib, silencing and activation of TUSC3, and xenografts. Were effects on bulk glioblastoma cells and glioblastoma cell lines checked?
Author Response
This manuscript entitled "Epigenetic activation of TUSC3 sensitizes glioblastoma to temozolomide independent of MGMT promoter methylation status." shows that treatment with 5-Aza sensitized to TMZ via upregulation of TUSC3 in MGMT-methylated glioma stem cells, and combination of 5-Aza and a MGMT inhibitor sensitized to TMZ in MGMT-unmethylated GSCs. This article is of value in the point that it may be provide a novel adjuvant treatment modality for TMZ-resistant glioblastoma. This reviewer has two concerns before recommendation for publication.
R: Thanks very much for the reviewer’s comments and we addressed the questions and concerns point by point below.
1. Figure 3, Because 5-Aza is a DNA methylation inhibitor, and MGMT is also influenced by its treatment, immunoblots of MGMT should be indicated in Fig 3A and 3C.
R: We thank the reviewer for the comments. In our study, as a DNA methyltransferase inhibitor, 5-Aza failed to alter MGMT promoter methylation status and further showed no impact on MGMT expression (Fig. S5). In Fig.3, we focused on the impact of TUSC3 expression (overexpression and knockout) on TMZ sensitivity, and Fig.3A&C here were only to validate TUSC3 modulation efficacy (overexpression and knockout).
2. GSCs were used in experiments of 5-Aza treatments with/without Lomeguatrib, silencing and activation of TUSC3, and xenografts. Were effects on bulk glioblastoma cells and glioblastoma cell lines checked?
R: We thank the reviewer for this valuable suggestion. In this revision, we have assessed 5-Aza’s impact on GBM cell lines including U251 and T98G. U251 is a GBM cell line which has hypermethylated MGMT, while T98G is a GBM cell line with hypomethylated MGMT [46]. We added new data in Fig.S8 which indicated that, in U251, 5-Aza dramatically activated TUSC3 expression and further increase TMZ sensitivity; while combination of 5-Aza and Lomeguatrib are required to activate TUSC3 and sensitize T98G cells to TMZ treatment, line 244-249. It should however be noted the GSCs that we employ in these studies are more reflective of GBM therapeutic resistance compared to GBM cell lines.
Reviewer 3 Report
In this study, the authors showed that 5-Aza treatment can sensitize MGMT-M GBM but not MGMT-UM GBM to TMZ therapy. Although previous studies have shown that the responsiveness of GBM to TMZ was dependent on the hypermethylation status of the MGMT gene promoter, the authors revealed that the sensitizing effect of 5-Aza was likely independent of the DNA methylation status of MGMT gene promoter. Through siRNA screening of a panel of differential expressed gene (the unique 5-Aza-upregulated genes in MGMT-M), they identified TUSC3 as a major determinant of 5-Aza induced TMZ sensitivity. Moreover, they showed that a combination of TUSC3 O/E and MGMT inhibitors can confer the TMZ sensitivity on MGMT-UM GBM cells. Thus, their work reported a new epigenetic mechanism of TMZ sensitivity in GBM. However, there are still a few concerns should be addressed.
Major points:
1. According to their findings, TUSC3 is likely a downstream effector of 5-Aza treatment which confer the TMZ responsiveness on GBM. However, little is known about how TUSC3 gene exerts a tumor suppressor function in GBM. The authors should provide more information on this point.
2. It is a little weird that 5-Aza treatment cannot demethylate the hypermethylated promoter and activate the expression of MGMT gene in MGMT-M GBM cells. The authors should discuss the possible mechanisms underlying this observation.
3. The relationship between TUSC3 and MGMT should also be discussed in depth.
Minor points:
1. Detailed information of drug treatment (i.e., how long the cells were treated by 5-Aza?) should be provided in methods or figure legends.
2. MGMG-UM ( a typo of MGMT-UM?) was used in several places.
Author Response
In this study, the authors showed that 5-Aza treatment can sensitize MGMT-M GBM but not MGMT-UM GBM to TMZ therapy. Although previous studies have shown that the responsiveness of GBM to TMZ was dependent on the hypermethylation status of the MGMT gene promoter, the authors revealed that the sensitizing effect of 5-Aza was likely independent of the DNA methylation status of MGMT gene promoter. Through siRNA screening of a panel of differential expressed gene (the unique 5-Aza-upregulated genes in MGMT-M), they identified TUSC3 as a major determinant of 5-Aza induced TMZ sensitivity. Moreover, they showed that a combination of TUSC3 O/E and MGMT inhibitors can confer the TMZ sensitivity on MGMT-UM GBM cells. Thus, their work reported a new epigenetic mechanism of TMZ sensitivity in GBM. However, there are still a few concerns should be addressed.
R: Thanks very much for the reviewer’s comments and we addressed the questions and concerns point by point below.
Major points:
1. According to their findings, TUSC3 is likely a downstream effector of 5-Aza treatment which confer the TMZ responsiveness on GBM. However, little is known about how TUSC3 gene exerts a tumor suppressor function in GBM. The authors should provide more information on this point.
R: We thank the reviewer for the suggestion. It has been previously reported that there is a significant decrease in TUSC3 expression in glioma tissues compared with normal adjacent tissue. Hence lower TUSC3 expression is associated with higher malignancy. Further, in vitro study revealed that TUSC3 upregulation in GBM cell lines suppressed cell proliferation and invasion by inhibition of Akt signaling pathway. Unfortunately, so far, little is known about how TUSC3 exerts a tumor suppressor gene function in the aggressive GBM stem niche, therefore further study regarding TUSC3 downstream mechanism is worth pursuing in our future study. We added some more introduction and discussion regarding this point in the revision line 94-100; 394-396. Meanwhile, our study also highlights the reprogramming role of TUSC3 activation in overcoming MGMT mediated resistance and focuses on providing prospects for clinical translation through pharmacological epigenetic reactivation.
2. It is a little weird that 5-Aza treatment cannot demethylate the hypermethylated promoter and activate the expression of MGMT gene in MGMT-M GBM cells. The authors should discuss the possible mechanisms underlying this observation.
R: We thank reviewer for the comments. It was noted that although 5-Aza is a DNA methyltransferase inhibitor, it cannot demethylated the hypermethylated MGMT promoter and activate the expression of MGMT in MGMT-M cells. Further 5-Aza sensitized MGMT-M GBMs to temozolomide (TMZ) which would not occur if 5-Aza demethylated the MGMT promoter leading to increased MGMT expression. It is therefore likely that 5-Aza is not an inhibitor for the DNA methyltransferase that maintains the methylation status of MGMT. This suggests that different DNMTs functionally maintain TUSC3 and MGMT promoter methylation status, and 5-Aza impacts on the DNMT effective on TUSC3 promoter but not MGMT. Meanwhile, DNA sequencing and quantification of mRNA and protein level revealed that MGMT activity was not always correlated with methylation of the core MGMT promoter (please see reference-39). Therefore, further studies are needed to address this phenomenon. We added these points in the “Discussion” section, line 333-345.
3. The relationship between TUSC3 and MGMT should also be discussed in depth.
R: We appreciate the reviewer for this great suggestion. According to our results, it appears that 5-Aza alone was sufficient in successfully reactivating TUSC3 in MGMT-M GSCs, while in MGMT-UM GSCs both 5-Aza and Lomeguatrib were required to reactivate TUSC3. This observation suggests that MGMT may play a critical role in suppression of TUSC3 transcriptional activation through an epigenetic way. Specifically, MGMT may inhibit demethylation of TUSC3 promoter when DNMT1 was suppressed by 5-Aza. Given that MGMT is not a known repressor of gene expression, it is possible that the suppression of TUSC3 promoter activation in MGMT-UM GSCs occurs through a complex or machinery that mediated by MGMT, which suggests a novel role of MGMT in cancers. We added those hypotheses and discussion in the “Discussion” section now, line 346-354.
Minor points:
1. Detailed information of drug treatment (i.e., how long the cells were treated by 5-Aza?) should be provided in methods or figure legends.
R: We appreciate this great suggestion. We have added the detail information regarding drug treatment in different assays in “Methods” section and “Figure legends”.
2. MGMG-UM (a typo of MGMT-UM?) was used in several places.
R: We thank the reviewer for pointing out these typos. We have rectified them.
Round 2
Reviewer 2 Report
The manuscript has been improved.